# DFITE: ESTIMATION OF INDIVIDUAL TREATMENT EFFECT USING DIFFUSION MODEL

## ABSTRACT

Learning individualized treatment effects (ITE) from observational data is a challenging task due to the existence of unobserved confounders. Previous methods mostly focus on assuming the Ignorability assumption ignoring the unobserved confounders or overlooking the impact of an apriori knowledge on the generation process of the latent variable, which can be quite impractical in real-world scenarios. Motivated by the recent advances in the latent variable modeling, we propose to capture the unobserved latent space using diffusion model, and accordingly to estimate the causal effect. More concretely, we build on the reverse diffusion process for the unobserved confounders as a Markov chain conditioned on an apriori knowledge. In order to implement our model in a feasible way, we derive the variational bound in closed form. In the experiments, we compare our model with the state-of-the-art methods based on both synthetic and benchmark datasets, where we can empirically demonstrate consistent improvements of our model on $\sqrt{\epsilon_{PEHE}}$ and $\epsilon_{ATE}$, respectively. To benefit this research direction, we release our project at https://github-dfite.github.io/dfite/.

## 1 INTRODUCTION

Estimating the Individual Treatment Effect (ITE) from observational data is a fundamental problem across a wide variety of domains. For example, re-weighting the training instances with the inverse propensity score (IPS) in recommender system Wang et al. (2021; 2022), measuring the effect of a certain medicine against a disease in healthcare Shalit (2020) and providing counterfactual visual explanations in computer vision Goyal et al. (2019). In this paper, we focus on these measure problems from confounders perspective.

How to measure the confounder is an essential problem in estimating ITE of an treatment $A$ (e.g.,medicine) on an individual with features $X$ (e.g., demographic characteristics ). A confounder is a variable which affects both the treatment and the outcome. On the one hand, one can account for ITE by controlling it with the Ignorability assumption in mind, i.e., there does not exists the unobserved confounder. The most crucial mechanism lie in balance the distribution among groups, usually through inverse propensity weighting (IPW) or covariate adjustment Yao et al. (2021); Louizos et al. (2017). While quite a lot of promising models have been proposed and achieved impressive performance, such as, the representative CFR Shalit et al. (2017), the augmented IPW estimator DR Funk et al. (2011) and so on, these methods build on the Ignorability assumption, which can be impractical in real-world scenarios. On the other hand, exactly collecting all of valid confounders is impossible in the general case. For example, demographic characteristics and genetic factor can both affect the choice of medication to a patient, and the patient's health. However we can only have access to the former in the observational data. As illustrated in Figure 1, the genetic factor acts as an unobserved confounder $Z$ both affecting the treatment $A$ and health outcomes $Y$, and without controlling it we can not block the backdoor path: $A \leftarrow Z \rightarrow Y$ as of estimating the causal effect of treatments on health measures.

In the past few years, some prominent generative models have been proposed to generate such unobserved confounder that we could utilize it to isolate the causal effect of treatment on outcome. For instance, VAE-based method CEVAE Louizos et al. (2017) assume that there exists a proxy variable in causal graph, and then generates the hidden confounder $Z$ by optimizing the variational lower bound of this graphical model, GANITE Yoon et al. (2018) aims to generate the counterfactual

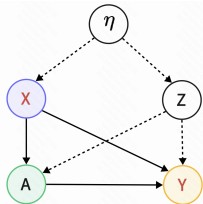

Figure 1: Motivating example on the generation process of the unobserved confounders. $A$ is a treatment, $X$ is an observed confounder, $Y$ denotes an outcome, $Z$ is an unobserved confounders and $\eta$ represents the common prior variable of $X$ and $Z$.

distributions using GAN, and accordingly to infer the ITE in an unbiased settings. Additionally, some advanced techniques are also applied to reconstruct or generate the hidden confounder, like Gaussian Processes Witty et al. (2020), Imitation Learning Zhang et al. (2020), deep latent variable models Josse et al. (2020) and more Li & Zhu (2022); Yao et al. (2021).

While great success has been made, these methods have some intrinsic limitations for modeling hidden confounder. For examples, GAN-based methods could be unstable in modeling ITE due to the adversarial losses. VAEs make substantially weaker assumptions in generating the structure of the hidden confounders Louizos et al. (2017), which could restrict the model's flexibility. In order to address these challenges, in this paper, we propose to generate the unobserved confounders using diffusion model. We aim to exploit two types of generation process called forward diffusion process and reverse diffusion process, respectively. The former process converts the observed confounders to a simple noise distribution by adding noise at each time step, in which the useful decomposition information can be preserved in transition kernel. And then the transition kernel are utilized to the unobserved confounder generation. The latter process are regarded as a Markov chain which is responsible for converting the noise distribution to our target latent distribution. By integrating these two processes, we can learn its transition kernel and accordingly reconstruct the desired unobserved confounders. Furthermore, we also design a generation factor as the condition for learning the transition kernel. The generation factor follows a prior distribution in our setting of generation. As illustrated in Figure 1, we assume that the generation factor $\eta$ can simultaneously affect the generation process of the observed confounder and the unobserved confounder. For examples, the environment in which the patient live and work can both affect the patient socio-economic status and gene for a certain disease. Therefore, the environment can be regarded as a generation factor, which plays a significant role in generating unobserved confounders.

The main contributions of this paper can be concluded as follows: (1) We propose to solve the task of unobserved confounders in causal inference with the diffusion model. (2) To realize the above idea, we first derive a variational lower bound of the likelihoodof the unobserved confounders conditional on the generation factor, and then reformulate that bound into a tractable expression in closed form. (3) We verify the effectiveness and generality of our framework by comparing with 12 state-of-the art methods on four datasets. The empirical studies manifest that the proposed method can achieve competitive gains both on synthetic and benchmark datasets.

## 2 PROBLEM FORMULATION

Under the Neyman-Rubin potential outcomes framework Rubin (2005), ITE estimation aims to measure the causal effect of a treatment or intervention $a \in \mathcal{A}$ on the outcome $y \in \mathcal{Y}$ for given the unit's confounders or descriptions $x \in \mathcal{X}$. Throughout this paper, we only focus on the binary treatment case, where $\mathcal{A} = \{0, 1\}$, $y$ represents the factual outcome. We treat units which received treatment, i.e., $a = 1$ as treated units and the other units with $a = 0$ as control units. The Individual Treatment Effect (ITE), also known as Conditional Average Treatment Effect (CATE) for unit $x$ is Shalit et al. (2017):

$$\tau(x) := \mathbb{E}[Y_1 - Y_0 | x] \tag{1}$$

Where $Y_a$ denotes the potential outcome for treatment $a$. In practice, we can only observe the factual outcome with respect to treatment assignment, i.e., $y = Y_0$ if $a = 0$, otherwise $y = Y_1$. Usually, we build on three significant assumptions to guarantee that the potential outcomes are identifiable from observational study.

**Assumption 1.** *Consistency. For a given patient with treatment assignment $a$, then the potential outcome for the treatment $a$ is the same as the observed (factual) outcome: $Y_a = y$*

**Assumption 2.** *Positivity (Overlap) . if $P(X = x) \neq 0$, then $P(A = a|X = x) > 0, \quad \forall a$ and $x$.*

**Assumption 3.** *Strong ignorability. For a given patient $(i)$, the treatment are independent of the potential outcomes if given the confounders $X$ : $A \perp\!\!\!\perp Y_1, Y_0|X$.*

With these assumptions in mind, the estimation on potential outcomes could be transformed into identifiable estimation from a statistical point of view. In other words, we call that the counterfactual outcomes can be identified under these assumptions, i.e, $\tau(x) = \mathbb{E}[Y|X = x, A = 1] - \mathbb{E}[Y|X = x, A = 0]$. From machine learning perspective, these observational dataset can be modeled via a standard supervised learning model, such as SVM, for estimating $\tau(x)$. However, this model could be unreliable and unviable employed to estimate the future counterfactual outcomes under the fact that without adjusting for the bias introduced by the unobserved confounders and imbalanced distribution between treated groups and control groups. The existing generative-based models can achieve promising results in generating unobserved confounders Louizos et al. (2017) and counterfactuals Yoon et al. (2018), which indeed eliminate the influence from backdoor between treatment and outcome. However, they have some inherent limitations, which would hinder the model's flexibility and performance. In this paper, we build on the prominent diffusion model to generate the unobserved confounders, and accordingly align the distribution between treated groups and control groups and measure the ITE. We proceed in two steps: (1) Generate the unobserved confounders conditioned on generation factor; (2) Balance the confounder's representation in latent space and measuring the ITE based on the observed and unobserved confounders.

## 3 DIFFUSION MODEL FOR UNOBSERVED CONFOUNDERS

In this section, we first formulate the diffusion model of both the forward and the reverse diffusion processes for unobserved confounders. Then, we reformulate the objective as a variational bound in closed form for training the model. In this end, we present our algorithm for estimating ITE. The implementation of the model is provided in the experiment section.

### 3.1 FORMULATION

In the observational study, we can only observe a set of full observational data with factual outcomes. In this case, we formulate the problem as follows: Let $D = \{(\boldsymbol{x}_i, a_i, y_i)\}_{i=1}^m$ denote the collected datasets, in which $m$ is the total number of observational data, $i$ is i-th unit with observed confounder $\boldsymbol{x}_i$, $a_i$ and $y_i$ are its corresponding treatment and factual outcome, respectively. We assume that each point $\boldsymbol{x}_i$ are sampled independently from a certain distribution, which we denote as $q(\boldsymbol{x}_i|\eta)$. As discussed in the introduction, $\eta$ is regarded as generation factor that affects the generation process. Since the observed confounders $\{\boldsymbol{x}_i\}_{i=1}^m$ are regarded as the initial samples in diffusion model, we add a superscript $(0)$ to it to indicate the time mark, i.e. $\{\boldsymbol{x}_i^{(0)}\}_{i=1}^m$. Additionally, for the sake of brevity, we ignore subscripts of characters unless otherwise specified. The forward diffusion process aims to converts the initial distribution into a noise distribution. Formally, the forward diffusion process is defined as a Markov chain Luo & Hu (2021); Ho et al. (2020):

$$q(\boldsymbol{x}^{(1:T)}|\boldsymbol{x}^{(0)}) = \prod_{t=1}^T q(\boldsymbol{x}^{(t)}|\boldsymbol{x}^{(t-1)}) \tag{2}$$

where $q(\boldsymbol{x}^{(t)}|\boldsymbol{x}^{(t-1)})$ is the handcraft transition kernel. At the previous time step, the kernel are responsible for adding noise to the points and at the next time step, which capable of modeling the distribution of points. One typical design for the transition kernel is Gaussian perturbation Yang et al. (2022); Luo & Hu (2021):

$$q(\boldsymbol{x}^{(t)}|\boldsymbol{x}^{(t-1)}) = \mathcal{N}(x^{(t)}; \sqrt{1 - \beta_t}x^{(t-1)}, \beta_t \boldsymbol{I}) \tag{3}$$

where $\beta_t \in (0, 1)$ is a hyper-parameter that control the rate of the forward diffusion process. More specifically, with $\alpha_t = 1 - \beta_t$ and $\bar{\alpha} = \prod_{i=1}^t \alpha_i$, we have:

$$q(\boldsymbol{x}^{(t)}|\boldsymbol{x}^{(t-1)}) = \mathcal{N}(\boldsymbol{x}^{(t)}; \sqrt{\bar{\alpha}_t}\boldsymbol{x}^{(0)}, (1 - \bar{\alpha})\boldsymbol{I}) \tag{4}$$

Through the above parameterization trick, we can easily obtain a sample of $x^{(t)}$ with noise for given a observed confounders $\boldsymbol{x}^{(0)}$:

$$\boldsymbol{x}^{(t)} = \sqrt{\bar{\alpha}_t}\boldsymbol{x}^{(0)} + \sqrt{1 - \bar{\alpha}_t}\boldsymbol{\epsilon_t}, \ \ \boldsymbol{\epsilon_t} \sim \mathcal{N}(\boldsymbol{0}, \boldsymbol{I}) \tag{5}$$

Our goal is to generate unobserved confounders with a meaningful generation factor encoded by the latent $\eta$. In our generation process, the reverse diffusion is capable of approximating the $q(\boldsymbol{x}_i^{(T)})$ from a simple noise distribution $p(\boldsymbol{x}_i^{(T)})$ that are given as the input. Therefore, with the latent representation $\eta$ and the preserved information from forward diffusion process, we can generate the desired unobserved confounders through the reverse Markov chain. Formally, the reverse diffusion process for generating unobserved confounders is:

$$p_{\boldsymbol{\theta}}(\boldsymbol{x}^{(0:T)}|\boldsymbol{\eta}) = p(\boldsymbol{x}^{(T)})\prod_{t=1}^{T} p_{\boldsymbol{\theta}}(\boldsymbol{x}^{(t-1)}|\boldsymbol{x}^{(t)}, \boldsymbol{\eta}) \tag{6}$$

where $p_{\boldsymbol{\theta}}(\boldsymbol{x}^{(t-1)}|\boldsymbol{x}^{(t)}, \boldsymbol{\eta})$ is learnable transition kernel and $\boldsymbol{\theta}$ is the model parameters. The learnable transition kernel takes the form of

$$p_{\boldsymbol{\theta}}(\boldsymbol{x}^{(t-1)}|\boldsymbol{x}^{(t)}, \boldsymbol{\eta}) = \mathcal{N}(\boldsymbol{x}^{(t-1)}; \mu_{\boldsymbol{\theta}}(\boldsymbol{x}^{(t)}, t, \boldsymbol{\eta}), \beta_t \boldsymbol{I})) \tag{7}$$

where the mean $\mu_{\boldsymbol{\theta}}(\boldsymbol{x}^{(t)}, t, \boldsymbol{\eta})$ are parameterized by deep neural networks learned in the optimization process and $\eta$ is the latent representation encoding the generation factor. In practice, we treat the noise distribution $p(\boldsymbol{x}_i^{(T)})$ as a standard normal distribution $\mathcal{N}(0, \boldsymbol{I})$. By applying the reverse Markov chain which given the generation factor and starting distribution $p(\boldsymbol{x}_i^{(T)})$, we can obtain the unobserved confounders with target distribution.

## 3.2 VARIATIONAL LOWER BOUND

With the formulated forward and reverse diffusion processes for unobserved confounders in mind, we now aims to formalize the training objective. Since directly optimizing the exact log-likelihood is intractable, we instead maximize its variational lower bound (VLB)(the detailed derivation is present in the Appendix):

$$\mathbb{E}[-\log p_{\boldsymbol{\theta}}(\boldsymbol{x}^{(0)})] \leq \underbrace{E_q\left[\log \frac{q(\boldsymbol{x}^{(1:T)}, \boldsymbol{\eta}|\boldsymbol{x}^{(0)})}{p_{\boldsymbol{\theta}}(\boldsymbol{x}^{(0:T)}, \boldsymbol{\eta}))}\right]}_{VLB} \tag{8}$$

Where $L_{VLB}$ is a common objective for training probabilistic generative models Luo & Hu (2021); Ho et al. (2020). We can further derive the $L_{VLB}$ as:

$$\begin{aligned}
L_{VLB} &= E_q\left[\log \frac{q(\boldsymbol{x}^{(1:T)}, \boldsymbol{\eta}|\boldsymbol{x}^{(0)})}{p_{\boldsymbol{\theta}}(\boldsymbol{x}^{(0:T)}, \boldsymbol{\eta}))}\right] \\
&= E_q\left[\sum_{t=2}^{T} D_{KL}\left(\underbrace{q(\boldsymbol{x}^{(t-1)}|\boldsymbol{x}^{(t)}, \boldsymbol{x}^{(0)})}_{A} \middle|\middle| \underbrace{p_{\boldsymbol{\theta}}(\boldsymbol{x}^{(t-1)}|\boldsymbol{x}^{(t)}, \boldsymbol{\eta})}_{B}\right) \right. \\
&\quad \left. - \log \underbrace{p_{\boldsymbol{\theta}}(\boldsymbol{x}^{(0)}|\boldsymbol{x}^{(1)}, \boldsymbol{\eta})}_{C} + D_{KL}\left(\underbrace{q_{\boldsymbol{\varphi}}(\boldsymbol{\eta}|\boldsymbol{x}^{(0)})}_{D} \middle|\middle| \underbrace{p(\boldsymbol{\eta})}_{E}\right)\right]
\end{aligned} \tag{9}$$

The above training objective can be optimized efficiently since each term in this objective is tractable. In order to make the objective more clear, we elaborate on the terms as following:

A $q(\boldsymbol{x}^{(t-1)}|\boldsymbol{x}^{(t)}, \boldsymbol{x}^{(0)})$ is usually computed by a closed-form Gaussian Luo & Hu (2021); Ho et al. (2020):

$$q(\boldsymbol{x}^{(t-1)}|\boldsymbol{x}^{(t)}, \boldsymbol{x}^{(0)}) = \mathcal{N}(\boldsymbol{x}^{(t-1)}; \boldsymbol{\mu}_t(\boldsymbol{x}^{(t)}, \boldsymbol{x}^{(0)}), \gamma t \boldsymbol{I}) \tag{10}$$

Where $\boldsymbol{\mu}_t(\boldsymbol{x}^{(t)}, \boldsymbol{x}^{(0)}) = \frac{\sqrt{\bar{a}_{t-1}}\beta_t}{1-\bar{a}_t}\boldsymbol{x}^{(0)} + \frac{\sqrt{a_t}(1-\bar{a}_{t-1})}{1-\bar{a}_t}\boldsymbol{x}^{(t)}$ and $\gamma_t = \frac{1-\bar{a}_{t-1}}{1-\bar{a}_t}\beta_t$.

B, C $p_{\boldsymbol{\theta}}(\boldsymbol{x}^{(t-1)}|\boldsymbol{x}^{(t)}, \boldsymbol{\eta})$ where $t \in \{1, 2, ..., T\}$ are trainable Gaussian distribution shown in Eq. 7. D $q_{\boldsymbol{\varphi}}(\boldsymbol{\eta}|\boldsymbol{x}^{(0)})$ are learnable posterior distribution, which aims to encode the input observed confounders $\boldsymbol{x}^{(0)}$ into the distribution of the latent generation factor $\eta$. Usually, we define it as: $q_{\boldsymbol{\varphi}}(\boldsymbol{\eta}|\boldsymbol{x}^{(0)}) = \mathcal{N}(\eta; \boldsymbol{\mu}_{\boldsymbol{\varphi}}(\boldsymbol{x}^{(0)}), \sum_{\boldsymbol{\varphi}}(\boldsymbol{x}^{(0)}))$. The last term E $p(\boldsymbol{\eta})$ is the prior distribution defined as isotropic

Gaussian $\mathcal{N}(0, \boldsymbol{I})$, which is the most common choice for approximating the target distribution. In the next section, we will show how to optimize this objective for generating the desired unobserved confounders and accordingly estimating ITE.

### 3.3 ALGORITHM FOR ESTIMATING ITE

The generation processs in the previous section lay the foundation for accurate ITE estimation tasks. In this section, we first give the widely used definitions in estimating ITE, and then present our methods.

Let $\Phi : \mathcal{X} \to \mathcal{R}$ be a representation function, $f : \mathcal{R} \times \{0, 1\} \to \mathcal{Y}$ be an hypothesis predicting the outcome of a patient's confounders $x$ given the representation confounders $\Phi(\boldsymbol{x})$ and the treatment assignment $a$. Let $L : \mathcal{Y} \times \mathcal{Y} \to \mathbb{R}_+$ be a loss function. The estimation of ITE by an hypothesis $f$ and a representation function $\Phi$ is:

$$\hat{\tau}(\boldsymbol{x}) = f(\Phi(\boldsymbol{x}), 1) - f(\Phi(\boldsymbol{x}), 0) \tag{11}$$

We utilize the expected Precision in Estimation of Heterogeneous Effect (PEHE) Hill (2011) to train our model. We define it as following:

$$\epsilon_{PEHE}(f) = \int_{\mathcal{X}} (\hat{\tau}(\boldsymbol{x}) - \tau(\boldsymbol{x}))^2 p(\boldsymbol{x}) dx \tag{12}$$

Based on the above analysis, we propose a method called DFITE ( Estimation of Individual Treatment Effect Using Diffusion Model), which take into account the unobserved confounders to estimate the ITE.

The optimization problem in our framework is shown as the following:

$$\min_{f, \Phi, \theta, \varphi} \sum_{i=1}^{m} w_i \cdot L(y_i, f(\Phi(\boldsymbol{x_i}, \boldsymbol{z}_i), a_i)) + L_{VLB}(\boldsymbol{x_i}) + \alpha \cdot \text{IPM}_G(\hat{p}_\Phi^{a=1}, \hat{p}_\Phi^{a=0}) \tag{13}$$

where $w_i$ is used to compensates for the difference in treatment group size. It can be calculated be the proportion of treated units in the population, the loss funcation $L$ is PEHE. the unobserved confounder $z_i$ is derived by diffusion model, i.e., $\boldsymbol{z}_i \sim \mu_{\boldsymbol{\theta}}(\boldsymbol{c}, t, \eta_i) + \beta_t \epsilon$ where $\epsilon, c \sim \mathcal{N}(0, I)$, $t$ is the time step in reverse Markov chain and $q_{\boldsymbol{\varphi}}(\eta_i | \boldsymbol{x_i})$ is the generation factor. Here, we use reparameterization trick to make the generation process more feasible. $L_{VLB}$ is the VLB loss that aims to learn the transition kernel. In practice, optimizing $L_{VLB}$ in our main objective is still a challenging task, since it requires to sum the expectation of the KL terms on all time steps. To make the training more efficient, we adopt the works in Ho et al. (2020) randomly choosing one term to optimize at each training step. The detailed training algorithm is present in Appendix. $\hat{p}_\Phi^{t=1}$ and $\hat{p}_\Phi^{t=0}$ are learned high-dimensional representation for treated and control groups respectively, $\text{IPM}_G(\cdot, \cdot)$ is the (empirical) integral probability metric w.r.t. a function family $G$. We adopt it to balance the treated and control distribution. The imbalance penalty $\alpha$ are used to weight the magnitude of the two distribution.

Based on above optimization, we can generate the latent confounders that are affected by a meaningful generation factor from a noise distribution and accordingly obtain the accurate ITE estimation.

We refer to the model minimizing equation 13 with the observed and unobserved confounders as DFITE. The model are trained by the adaptive moment estimation (Adam) Kingma & Ba (2014). The details are described in the Appendix.

## 4 IDENTIFYING INDIVIDUAL TREATMENT EFFECT

Our goal in this paper is to estimate the ITE. To do that, we assume that there exist the unobserved confounders that can both affect the treatment and outcome. As there exists the hidden confounder, we can not block the backdoor path $A \leftarrow Z \to Y$, which results in $p(Y_a | X = x, A = a) \neq p(Y | X = x, A = a)$. In order to make the potential outcomes identifiable, we derive following theory:

**Theorem 1.** *If we can recover the latent confounders distribution $p(Z)$, then we can identify the ITE.*

*Proof.* If we can make the potential outcome with $A = a$ identifiable, then the ITE is identical. With the definitions of potential outcomes, we have:

$$
\begin{aligned}
p(Y_a|A = a) &= \int_{Z \times X} p(Y_a|A = a, X, Z) p(X, Z|A = a) dZ dX \\
&\overset{(a)}{=} \int_{Z \times X} p(Y|A = a, X, Z) p(X, Z) dZ dX
\end{aligned}
\tag{14}
$$

$\square$

Where we use the independent conditions of $Y_a$ and $A = a$ for given the complete confounders $(X, Z)$ in (a). The theoretical results demonstrate that the potential outcome can be identified from the distribution $p(X, Z)$.

## 5 EXPERIMENTS

### 5.1 EXPERIMENT SETUP

ITE estimation is more difficult compared to machine learning tasks, the reason is that we rarely have access to ground-truth treatment effect in real-world scenario. In order to measure the accurate estimation of ITE, we conduct experiments based on two types of synthetic datasets and two standard benchmark datasets. The detailed description about these datasets are shown as follows:

**ACIC 2016.** This is a common benchmark dataset introduced by Dorie et al. (2019), which was developed for the 2016 Atlantic Causal Inference Conference competition data Dorie et al. (2019). It comprises 4,802 units (28% treated, 72% control) and 82 confounders measuring aspects of the linked birth and infant death data (LBIDD). The dataset are generated randomly according to the data generating process setting. We conduct experiments over randomly picked 100 realizations with 63/27/10 train/validation/test splits.

**IHDP.** Hill (2011) introduced a semi-synthetic dataset for causal effect estimation. The dataset was based on the Infant Health and Development Program (IHDP), in which the confounders were generated by a randomized experiment investigating the effect of home visits by specialists on future cognitive scores. it consists of 747 units(19% treated, 81% control ) and 25 confounders measuring the children and their mothers. Following the common settings in Qin et al. (2021); Shalit et al. (2017), We average over 1000 replications of the outcomes with 63/27/10 train/validation/test splits.

**Sim-$z$.** This synthetic dataset is based on observed and unobserved confounders that are both obtained from an normal Gaussian distribution. We adopt the generation process proposed in Assaad et al. (2021); Louizos et al. (2017) to simulate the treatment effect as:

$$
\begin{aligned}
&x_i \sim \mathcal{N}(0, \sigma_X^2); \quad z_i \sim \mathcal{N}(0.5, \sigma_Z^2); \\
&a_i|x_i, z_i \sim \text{Bernoulli}(\sigma(0.5x_i^T \beta_X + 0.5z_i^T \beta_Z)) \\
&\epsilon_i \sim \mathcal{N}(0, \sigma_Y^2); \quad \mathbf{y}_i(0) = x_i^T \beta_a + z_i^T \beta_b - r + \epsilon_i \\
&\mathbf{y}_i(1) = x_i^T \beta_a + z_i^T \beta_b + x_i^T \beta_c + z_i^T \beta_d + r + \epsilon_i
\end{aligned}
\tag{15}
$$

where $\sigma$ is the logistic sigmoid function. This generation process satisfies the assumptions of ignorability and positivity. We randomly construct 100 replications of such datasets with 10,000 units (50% treated, 50% control) and 50 confounders by setting $\sigma_X$ and $\sigma_Y$ both to 0.5, $\beta_T$, $\beta_0$ and $\beta_1$ are all sampled from $\mathcal{N}(0, 1)$.

**Sim-$\eta$.** This synthetic dataset aims to mimic the causal data generating process in terms of a prior distribution specified in advance. We simulate the treatment effect as:

$$
\begin{aligned}
&\eta_i \sim \mathcal{N}(0, I); \\
&x_i|\eta \sim \mathcal{N}(\eta_i, \sigma_{x_1}^2 \eta_i + \sigma_{x_0}^2 (1 - \eta_i)); \\
&z_i|\eta \sim \mathcal{N}(\eta_i + 0.5, \sigma_{z_1}^2 \eta_i + \sigma_{z_0}^2 (1 - \eta_i));
\end{aligned}
\tag{16}
$$

Table 1: Individual treatment effect estimation on ACIC, IHDP and two types of Sim datasets. The top module consists of baselines from recent works. The bottom module consists of our proposed method. In each module, we present each of the result with form mean ± standard deviation and we use bold fonts to label the best performance. Lower is better.

| Datasets | ACIC | | IHDP | | Sim-$z$ | | Sim-$\eta$ | |
|---|---|---|---|---|---|---|---|---|
| Metric | $\sqrt{\epsilon_{PEHE}}$ | $\epsilon_{ATE}$ | $\sqrt{\epsilon_{PEHE}}$ | $\epsilon_{ATE}$ | $\sqrt{\epsilon_{PEHE}}$ | $\epsilon_{ATE}$ | $\sqrt{\epsilon_{PEHE}}$ | $\epsilon_{ATE}$ |
| RF | $3.09 \pm 1.48$ | $1.16 \pm 1.40$ | $4.61 \pm 6.56$ | $0.70 \pm 1.50$ | $4.92 \pm 0.00$ | $0.61 \pm 0.01$ | $12.13 \pm 0.00$ | $3.21 \pm 0.02$ |
| CF | $1.86 \pm 0.73$ | $0.28 \pm 0.27$ | $4.46 \pm 6.53$ | $0.81 \pm 1.36$ | $4.70 \pm 0.00$ | $0.74 \pm 0.00$ | $6.96 \pm 0.00$ | $1.25 \pm 0.00$ |
| S-learner | $3.86 \pm 1.45$ | $0.41 \pm 0.35$ | $5.76 \pm 8.11$ | $0.96 \pm 1.80$ | $4.96 \pm 0.00$ | $0.84 \pm 0.00$ | $11.74 \pm 0.00$ | $0.92 \pm 0.00$ |
| T-learner | $2.33 \pm 0.86$ | $0.79 \pm 0.68$ | $4.38 \pm 7.85$ | $2.16 \pm 6.17$ | $5.68 \pm 0.08$ | $0.94 \pm 0.10$ | $6.87 \pm 0.12$ | $1.05 \pm 0.29$ |
| CEVAE | $5.63 \pm 1.58$ | $3.96 \pm 1.37$ | $7.87 \pm 7.41$ | $4.39 \pm 1.63$ | $5.20 \pm 0.03$ | $1.78 \pm 0.12$ | $12.83 \pm 0.61$ | $5.37 \pm 0.47$ |
| BNN | $2.00 \pm 0.86$ | $0.43 \pm 0.36$ | $3.17 \pm 3.72$ | $1.14 \pm 1.70$ | $5.09 \pm 0.04$ | $1.37 \pm 0.19$ | $12.49 \pm 0.21$ | $5.04 \pm 0.52$ |
| DragonNet | $1.26 \pm 0.32$ | $0.15 \pm 0.13$ | $1.46 \pm 1.52$ | $0.28 \pm 0.35$ | $4.09 \pm 0.10$ | $0.50 \pm 0.32$ | $\mathbf{6.16 \pm 0.10}$ | $0.47 \pm 0.30$ |
| TARNet | $1.30 \pm 0.46$ | $\mathbf{0.15 \pm 0.12}$ | $1.49 \pm 1.56$ | $0.29 \pm 0.40$ | $4.10 \pm 0.11$ | $0.52 \pm 0.34$ | $\mathbf{6.16 \pm 0.10}$ | $0.44 \pm 0.36$ |
| GANITE | $4.27 \pm 1.34$ | $3.27 \pm 1.37$ | $6.79 \pm 5.60$ | $4.43 \pm 1.43$ | $4.07 \pm 0.06$ | $1.92 \pm 0.09$ | $10.78 \pm 0.15$ | $5.83 \pm 0.20$ |
| CFR$_{MMD}$ | $1.24 \pm 0.31$ | $0.17 \pm 0.14$ | $1.51 \pm 1.66$ | $0.30 \pm 0.52$ | $4.06 \pm 0.09$ | $\mathbf{0.40 \pm 0.32}$ | $6.16 \pm 0.11$ | $0.45 \pm 0.33$ |
| CFR$_{WASS}$ | $1.27 \pm 0.38$ | $0.15 \pm 0.12$ | $1.43 \pm 1.61$ | $0.27 \pm 0.41$ | $4.10 \pm 0.09$ | $0.52 \pm 0.36$ | $6.18 \pm 0.11$ | $0.49 \pm 0.35$ |
| QHTE | $1.32 \pm 0.41$ | $0.19 \pm 0.18$ | $1.83 \pm 1.90$ | $0.34 \pm 0.43$ | $6.05 \pm 0.23$ | $0.58 \pm 0.26$ | $7.39 \pm 0.38$ | $0.84 \pm 0.43$ |
| DFITE | $\mathbf{1.20 \pm 0.07}$ | $0.20 \pm 0.14$ | $\mathbf{0.59 \pm 0.08}$ | $\mathbf{0.17 \pm 0.11}$ | $\mathbf{4.05 \pm 0.08}$ | $0.41 \pm 0.3$ | $6.17 \pm 0.12$ | $\mathbf{0.44 \pm 0.34}$ |

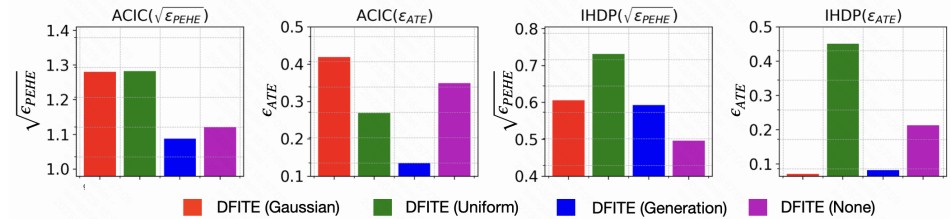

Figure 2: Performance comparison between our model and its variants on the unobserved confounders. The performances of different types of unobserved confounders are labeled with different colors. Lower is better.

We sample the generation factor $\eta$ from a standard normal distribution and accordingly generate the confounder $x$ and $z$ . The remaining generation process is the same as Sim-$\eta$. This generation process satisfies the conditions of Theorem 1. We randomly construct 100 replications of such datasets with 10,000 units (50% treated, 50% control) and 50 confounders by setting $\sigma_{x_1}^2, \sigma_{x_0}^2, \sigma_{z_1}^2$, and $\sigma_{z_0}^2$ to 0.5,0.3,0.7 and 0.9 respectively.

**Baselines.** We compare our model with the following 12 representative baselines: Random Forests (RF) Breiman (2001), Causal Forests (CF) Wager & Athey (2018), Causal Effect Variational Autoencoder (CEVAE) Louizos et al. (2017), DragonNet Shi et al. (2019), Meta-Learner algorithms S-Learner Nie & Wager (2021) and T-Learner Künzel et al. (2019), Balancing Neural Network (BNN) Johansson et al. (2016), Treatment-Agnostic Representation Network (TARNet) Shalit et al. (2017), Estimation of individualized treatment effects using generative adversarial nets (GANITE) Yoon et al. (2018) as well as CounterFactual Regression with the Wasserstein metric (CFR$_{WASS}$) Shalit et al. (2017) and the squared linear MMD metric (CFR$_{MMD}$) Shalit et al. (2017), along with a extension of CRF method Query-based Heterogeneous Treatment Effect estimation (QHTE) Qin et al. (2021).

**Implementation details.** We implement our methods based on QHTE Qin et al. (2021). We adopt the commonly used metrics including Rooted Precision in Estimation of Heterogeneous Effect (PEHE) Hill (2011) and Mean Absolute Error (ATE) Shalit et al. (2017) for evaluating the quality of ITE. Formally, they are defined as:

$$\sqrt{\epsilon_{PEHE}} = \sqrt{\frac{1}{n}\sum_{i=1}^{n}\left(\hat{\tau}_i - \tau_i\right)^2}, \epsilon_{ATE} = |\frac{1}{n}\sum_{i=1}^{n}(\hat{\tau}) - \frac{1}{n}\sum_{i=1}^{n}(\tau)| \tag{17}$$

Figure 3: t-SNE visualization of the balanced representations of ACIC learned by our algorithm DFITE with 4 types of unobserved confounders.

where $\hat{\tau}_i$ and $\tau_i$ stand for the predicted ITE and the ground truth ITE for the $i$-th instance respectively. The more details about the implementation of all adopted baselines and our methods and full experimental settings are presented in Appendix

## 5.2 OVERALL RESULTS

The overall comparison results are presented in Table 1, from which we can see: among the baselines, distance metric methods like $\text{CFR}_{WASS}$ and $\text{CFR}_{MMD}$, can obtain more performance gain both than the non-distance metric ones like GANITE and CEVAE, and traditional machine learning models like RF and CF, in most cases. This observation is consistent to our expectations and also agrees with the previous work Shalit et al. (2017), and verify that minimizing the distance between the treated and control groups on the studied latent space can effectively eliminate the distribution shift and lead to better performance on ITE estimation.

It is encouraging to see that our model DFITE can achieve the best performance on different datasets and evaluation metrics in more cases. The results verify the effectiveness of our idea. Comparing with the baselines, we take advantages of both the observed and unobserved confounders, which enable us to not only facilitate the identification of potential outcome, but also enhance to balance the studied representations between the treated and control groups. As a result, our model can always achieve the better performance on the estimation of ITE.

## 5.3 CONFOUNDERS CERTIFICATION

In this section, we would like to study whether different unobserved confounders in our model are necessary. To this end, we compare our model with four different unobserved confounders: DFITE(Gaussian) is a method with the unobserved confounders sampled randomly from the normal Gaussian $\mathcal{N}(0,1)$, DFITE(Uniform) is based on Uniform $\mathcal{U}(-1.5,1.5)$, DFITE(Generation) is our method, in which the unobserved confounder are generated by a reverse diffusion model and DFITE(None) is the typical representation methods with the ignorability assumption hold. Due to the space limitation, we present the results based on $\sqrt{\epsilon_{PEHE}}$ and $\epsilon_{ATE}$ and the datasets of ACIC and IHDP. From the results shown in Figure 2, we can see: DFITE(Gaussian) performs better than DFITE(Uniform). We speculate that the unobserved confounders sampled from normal Gaussian is more common than sampled from Uniform in practice. Nevertheless, both of which performs worse than DFITE(None). This maybe because by randomly drawing unprovable unobserved confounders, the ITE model are forced to encode the the noise samples, which result in a biased estimation. It is interesting to see that when we add the generated confounders in estimating ITE, the performance of DFITE(Generation) is better than DFITE(None) in more cases. This observation demonstrates the effectiveness of our idea on capturing the unobserved confounders.

## 5.4 LEARNED REPRESENTATIONS

In this section, we investigate the influence of different types of unobserved confounders in balancing the studied representations between treated and control groups, where the parameter settings follow the above experiments, and we compare the explanations generated by DFITE(Gaussian), DFITE(Uniform),DFITE(Generation) and DFITE(None). From the results shown in Figure 3 and Figure 4 we can see: all of these methods can perform several regions where the representations are indeed balanced. Such that they appear equal in studied high-dimension space. The results demonstrate that the distance metric used to balance two distributions play a significant role in

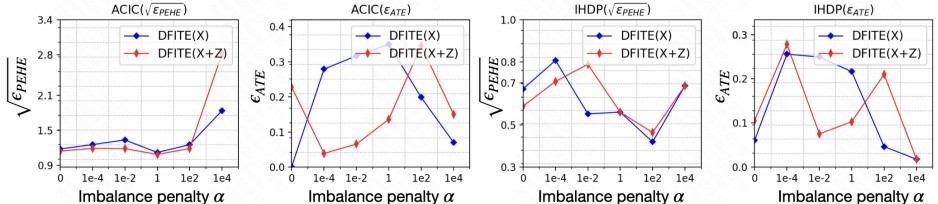

Figure 4: t-SNE visualization of the balanced representations of IHDP learned by our algorithm DFITE with 4 types of unobserved confounders.

Figure 5: Influence of the imbalance penalty $\alpha$ on our model performance in terms of $\sqrt{\epsilon_{PEHE}}$ and $\epsilon_{ATE}$. The performances of different types of confounders are labeled with different colors. Lower is better.

improving the estimation of ITE. Furthermore, in the illustration of representations generated by DFITE(Generation), we can find that some regions appear a strip-like representation on IHDP, whereas some regions appear rod-like shape on ACIC, where both of which have a smaller overlap. This observation demonstrate that the unobserved confounders generated by reverse diffusion model can contribute to balancing the studied distribution between treated and control groups.

## 5.5 PARAMETER STUDY

In this section, we analyze the influence of the key hyper parameters in our objective 13, we report the results on the same datasets and evaluation metric as the above experiments. The imbalance penalty $\alpha$ determines the magnitude of IPM in overall loss function. We tune $\alpha$ in $[0, 1e-4, 1e-2, 1, 1e2, 1e4]$. In order to investigate the influence of the unobserved confounders in parameter study, we compare our model with its two combinations of confounders: DFITE(X) is a model based on the observed confounder X and DFITE(X+Z) is based on both the observed and unobserved confounders X and Z, where $Z$ is generated by a reverse diffusion model. The results are presented in Figure 5, from which we can see: for both methods of DFITE(X) and DFITE(X+Z), the performance fluctuates a lot as $\alpha$ varies, but the best performance is usually achieved when $\alpha$ is moderate. This agrees with our expectation, i.e., too small $\alpha$ may lead to the imbalanced studied representation, while too large $\alpha$ may hinder the accurate estimation of ITE. Between DFITE(X) and DFITE(X+Z), we can find that the red line usually appears below blue line. The intuitive example suggests that the performance of DFITE(X+Z) tend to better than DFITE(X) as $\alpha$ varies. As expected, the unobserved confounders generated by our methods contributes to the estimation of ITE and should not be ignored.

## 6 CONCLUSION

In this paper, we propose to generate the unobserved confounders, and accordingly to facilitate the identification of potential outcome, as well as enhancing the learned representations. To achieve this goal, we first reconstruct the unobserved confounders by a reverse diffusion model, and then to estimation the ITE and balance the distribution between the treated and control groups based on the combination of the observed and unobserved confounders. In the experiments, we evaluate our framework based on both synthetic and real-world datasets to demonstrate its effectiveness and generality. This paper makes a first step on applying the idea of diffusion model to the field of estimating ITE. There is still much room for improvement. To begin with, one can incorporate different prior knowledge into the generation process, and at the same time devise effective mechanism for encouraging identification to causal inference. In addition, in order to reduce the time-consuming, people can also investigate the specific time step in generating the unobserved confounders.

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

## A APPENDIX

### A.1 RELATED WORK

**Estimating individual treatment effect.** Accurate and correct estimation of individual treatment effect is an challenging task in real-world scenarios, since the lack of counterfactuals can lead to an biased estimation from observational study. To alleviate this problem, early methods, like re-weighting

models Austin (2011); Imai & Ratkovic (2014); Fong et al. (2018), use the Inverse propensity weighting (IPW) mechanism to reduce selection bias based on covariates. Another active top line of research is to incorporate traditional machine learning into the study of estimating ITE, like Bayesian Additive Regression Trees (BART) Hill (2011), Random Forests (RF) Breiman (2001), Causal Forests (CF) Wager & Athey (2018), etc. In order to balance the distribution among groups in representation space, some advanced models are designed, like DragonNet Shi et al. (2019) , CFR Shalit et al. (2017), QHTE Qin et al. (2021),etc. There models use more flexibility and sophisticated technique, like Integral Probability Metric (IPM), to pull in that distributions and while minimize generalization bound for ITE estimation. While remarkable progresses have made by these models, the premise that they need to get the Ignorability assumption hold. However, the Ignorability assumption is untestable in practice. To this end, some promising deep generative models are proposed to generate latent variables. For example, Causal Effect Variational Autoencoder (CEVAE) Louizos et al. (2017) leverage Variational Autoencoders to obtain the unobserved confounders and simultaneously infer causal effects, GAITE Yoon et al. (2018) use Generative Adversarial Nets (GANs) framework to capture the uncertainty in the counterfactual distributions. While remarkable progresses have made by these models, here are some intrinsic limitations for modeling latent variables. For examples, GAN-based methods could be unstable in modeling ITE due to the adversarial losses. VAEs make substantially weaker assumptions in generating the structure of the hidden confounders Louizos et al. (2017), which could restrict the model's flexibility. In this paper, we build on diffusion model to generate the unobserved confounders and accordingly to measure ITE. The benefits are presented in two aspects; (1) Comparing to CEVAE, diffusion model has less assumptions in our settings, which is great of importance for estimating ITE; (2) Diffusion model has a comparative stable loss function, which indeed contribute to the generation process of unobserved confounders.

**Diffusion Model.** Diffusion Model is a concept describing the study of the deep generative process. It basically involves two types of Markov chains, called forward diffusion process and reverse diffusion process respectively. The former is capable of converting any data distribution into a simple or noise prior distribution, while the latter aims to reconstruct the original data distribution by a reverse Markov chain. In that process, the goal is to learn a transition kernels parameterized by deep neural networks Yang et al. (2022) and accordingly to generate the desired data. Due to its flexibility and strength, recent years have witnessed many studies on incorporating diffusion model into a variety of challenging domains Yang et al. (2022); Luo & Hu (2021); Ho et al. (2020) and achieved impressive results. For example, inspired by the diffusion model in computer vision, Luo & Hu (2021) proposes to generate 3D point cloud by a Markov chain conditioned on certain shape latent. In natural language processing, in order to handle more complex controls in generating text, Diffusion-LM Li et al. (2022) is proposed as a new language model based on continuous diffusion. Additionally, Adaptive Denoising Purification Yoon et al. (2021) proposes an effective randomized purification scheme to purify attacked images in robust learning. Similar to these applications, in this paper, we proposed to generate the unobserved confounders by a Markov chain conditioned on the generate factor that is derived from the observed confounders. To the best of our knowledge, this is the first work on estimating individual treatment effect.

## A.2 EXPERIMENT DETAILS.

We implement our methods based on QHTE Qin et al. (2021). We use the same set of hyperparameters for DFITE across four datasets. More precisely, we employ 3 similar fully-connected exponential-linear layers for the encoder $q_{\boldsymbol{\varphi}}(\boldsymbol{\eta}|\boldsymbol{x}^{(0)})$, the transition kernel $p_{\boldsymbol{\theta}}(\boldsymbol{x}^{(t-1)}|\boldsymbol{x}^{(t)}, \boldsymbol{\eta})$, representation function $\Phi$, and the ITE prediction function $f$ respectively. The difference is that layer sizes are 128 for both $q_{\boldsymbol{\varphi}}(\boldsymbol{\eta}|\boldsymbol{x}^{(0)})$ and $p_{\boldsymbol{\theta}}(\boldsymbol{x}^{(t-1)}|\boldsymbol{x}^{(t)}, \boldsymbol{\eta})$, 200 for $\Phi$, and 100 for $f$. we use Batch normalization Ioffe & Szegedy (2015) to facilitate training, and all but the output layer use ReLU (Rectified Linear Unit) Agarap (2018) as activation functions. In the main optimization objective, we set $\alpha$ and $\beta$ both to 1. We adopt the commonly used metrics including Rooted Precision in Estimation of Heterogeneous Effect (PEHE) Hill (2011) and Mean Absolute Error (ATE) Shalit et al. (2017) for evaluating the quality of ITE. Formally, they are defined as:

$$\sqrt{\epsilon_{PEHE}} = \sqrt{\frac{1}{n}\sum_{i=1}^{n}(\hat{\tau}_i - \tau_i)^2}, \quad \epsilon_{ATE} = |\frac{1}{n}\sum_{i=1}^{n}(\hat{\tau}) - \frac{1}{n}\sum_{i=1}^{n}(\tau)| \tag{18}$$

where $\hat{\tau}_i$ and $\tau_i$ stand for the predicted ITE and the ground truth ITE for the $i$-th instance respectively.

## B  DETAILED DERIVATIONS.

The variational lower bound (VLB)is :

$$\mathbb{E}[-\log p_{\boldsymbol{\theta}}(\boldsymbol{x}^{(0)})] \leq \underbrace{E_q\left[\log \frac{q(\boldsymbol{x}^{(1:T)}, \boldsymbol{\eta}|\boldsymbol{x}^{(0)})}{p_{\boldsymbol{\theta}}(\boldsymbol{x}^{(0:T)}, \boldsymbol{\eta}))}\right]}_{VLB} \tag{19}$$

*Proof.* We present the detailed derivations of the Negative Log-Likelihood in Eq. 19.

$$
\begin{aligned}
& -\log p_{\boldsymbol{\theta}}(\boldsymbol{x}^{(0)}) \\
& \leq \underbrace{-\log p_{\boldsymbol{\theta}}(\boldsymbol{x}^{(0)}) + D_{KL}(q(\boldsymbol{x}^{(1:T)}, \boldsymbol{\eta}|\boldsymbol{x}^{(0)})||p_{\boldsymbol{\theta}}(\boldsymbol{x}^{(1:T)}|\boldsymbol{x}^{(0)}, \boldsymbol{\eta}))}_{A} \\
& \leq \log p_{\boldsymbol{\theta}}(\boldsymbol{x}^{(0)}) + \underbrace{E_q\left[\log \frac{q(\boldsymbol{x}^{(1:T)}, \boldsymbol{\eta}|\boldsymbol{x}^{(0)})}{p_{\boldsymbol{\theta}}(\boldsymbol{x}^{(1:T)}|\boldsymbol{x}^{(0)}, \boldsymbol{\eta}))}\right]}_{B} \\
& \leq -\log p_{\boldsymbol{\theta}}(\boldsymbol{x}^{(0)}) + \underbrace{E_q\left[\log \frac{q(\boldsymbol{x}^{(1:T)}, \boldsymbol{\eta}|\boldsymbol{x}^{(0)})}{p_{\boldsymbol{\theta}}(\boldsymbol{x}^{(0:T)}, \boldsymbol{\eta}))}\right] + \log p_{\boldsymbol{\theta}}(\boldsymbol{x}^{(0)})}_{C} \\
& \leq \underbrace{E_q\left[\log \frac{q(\boldsymbol{x}^{(1:T)}, \boldsymbol{\eta}|\boldsymbol{x}^{(0)})}{p_{\boldsymbol{\theta}}(\boldsymbol{x}^{(0:T)}, \boldsymbol{\eta}))}\right]}_{VLB}
\end{aligned}
\tag{20}
$$

$\square$

We can further derive the $L_{VLB}$ as:

$$
\begin{aligned}
L_{VLB} &= E_q\left[\log \frac{q(\boldsymbol{x}^{(1:T)}, \boldsymbol{\eta}|\boldsymbol{x}^{(0)})}{p_{\boldsymbol{\theta}}(\boldsymbol{x}^{(0:T)}, \boldsymbol{\eta}))}\right] \\
&= E_q\left[\sum_{t=2}^{T} D_{KL}\left(\underbrace{q(\boldsymbol{x}^{(t-1)}|\boldsymbol{x}^{(t)}, \boldsymbol{x}^{(0)})}_{A} || \underbrace{p_{\boldsymbol{\theta}}(\boldsymbol{x}^{(t-1)}|\boldsymbol{x}^{(t)}, \boldsymbol{\eta})}_{B}\right)\right. \\
&\quad \left. -\log \underbrace{p_{\boldsymbol{\theta}}(\boldsymbol{x}^{(0)}|\boldsymbol{x}^{(1)}, \boldsymbol{\eta})}_{C} + D_{KL}\left(\underbrace{q_{\boldsymbol{\varphi}}(\boldsymbol{\eta}|\boldsymbol{x}^{(0)})}_{D} || \underbrace{p(\boldsymbol{\eta})}_{E}\right)\right]
\end{aligned}
\tag{21}
$$

Table 2: Statistics of the datasets used in our experiments.

| Dataset | #Replications | #Units | #confounders | Treated Ratio | Control Ratio |
|---------|---------------|--------|--------------|---------------|---------------|
| ACIC | 100 | 4,802 | 82 | 28% | 72% |
| IHDP | 1,000 | 747 | 25 | 19% | 81% |
| Sim-$z$ | 100 | 10,000 | 50 | 50% | 50% |
| Sim-$\eta$ | 100 | 10,000 | 50 | 50% | 50% |

*Proof.* We present the detailed derivations of the VLB in Eq. 21.

$$
L_{VLB} = E_q \left[ \log \frac{q(\boldsymbol{x}^{(1:T)}, \boldsymbol{\eta} | \boldsymbol{x}^{(0)})}{p_{\boldsymbol{\theta}}(\boldsymbol{x}^{(0:T)}, \boldsymbol{\eta}))} \right]
$$

$$
= E_q \left[ \log \frac{q(\boldsymbol{\eta} | \boldsymbol{x}^{(0)}) \prod_{t=1}^{T} q(\boldsymbol{x}^{(t)} | \boldsymbol{x}^{(t-1)})}{p_{\boldsymbol{\theta}}(\boldsymbol{\eta}) p(\boldsymbol{x}^{(T)}) \prod_{t=1}^{T} p_{\boldsymbol{\theta}}(\boldsymbol{x}^{(t-1)} | \boldsymbol{x}^{(t)}, \boldsymbol{\eta})} \right]
$$

$$
= E_q \left[ - \log p(\boldsymbol{x}^{(T)}) + \sum_{t=1}^{T} \log \frac{q(\boldsymbol{x}^{(t)} | \boldsymbol{x}^{(t-1)})}{p_{\boldsymbol{\theta}}(\boldsymbol{x}^{(t-1)} | \boldsymbol{x}^{(t)}, \boldsymbol{\eta})} + \log \frac{q_{\varphi}(\boldsymbol{\eta} | \boldsymbol{x}^{(0)})}{p_{\boldsymbol{\theta}}(\boldsymbol{\eta})} \right]
$$

$$
= E_q \left[ - \log p(\boldsymbol{x}^{(T)}) + \log \frac{q(\boldsymbol{x}^{(1)}) | \boldsymbol{x}^{(0)})}{p_{\boldsymbol{\theta}}(\boldsymbol{x}^{(0)} | \boldsymbol{x}^{(1)}), \boldsymbol{\eta})} + \sum_{t=2}^{T} \log \left( \frac{q(\boldsymbol{x}^{(t-1)} | \boldsymbol{x}^{(t)}, \boldsymbol{x}^{(0)})}{p_{\boldsymbol{\theta}}(\boldsymbol{x}^{(t-1)} | \boldsymbol{x}^{(t)}, \boldsymbol{\eta})} \cdot \frac{q(\boldsymbol{x}^{(t)} | \boldsymbol{x}^{(0)})}{q(\boldsymbol{x}^{(t-1)} | \boldsymbol{x}^{(0)})} \right) + \log \frac{q_{\varphi}(\boldsymbol{\eta} | \boldsymbol{x}^{(0)})}{p_{\boldsymbol{\theta}}(\boldsymbol{\eta})} \right]
$$

$$
= E_q \left[ - \log p(\boldsymbol{x}^{(T)}) + \log \frac{q(\boldsymbol{x}^{(1)}) | \boldsymbol{x}^{(0)})}{p_{\boldsymbol{\theta}}(\boldsymbol{x}^{(0)} | \boldsymbol{x}^{(1)}), \boldsymbol{\eta})} + \sum_{t=2}^{T} \log \frac{q(\boldsymbol{x}^{(t-1)} | \boldsymbol{x}^{(t)}, \boldsymbol{x}^{(0)})}{p_{\boldsymbol{\theta}}(\boldsymbol{x}^{(t-1)} | \boldsymbol{x}^{(t)}, \boldsymbol{\eta})} + \log \frac{q(\boldsymbol{x}^{(T)} | \boldsymbol{x}^{(0)})}{q(\boldsymbol{x}^{(1)} | \boldsymbol{x}^{(0)})} + \log \frac{q_{\varphi}(\boldsymbol{\eta} | \boldsymbol{x}^{(0)})}{p_{\boldsymbol{\theta}}(\boldsymbol{\eta})} \right]
$$

$$
= E_q \left[ - \log \frac{p(\boldsymbol{x}^{(T)})}{q(\boldsymbol{x}^{(T)} | \boldsymbol{x}^{(0)})} - \log p_{\boldsymbol{\theta}}(\boldsymbol{x}^{(0)} | \boldsymbol{x}^{(1)}), \boldsymbol{\eta}) + \sum_{t=2}^{T} \log \frac{q(\boldsymbol{x}^{(t-1)} | \boldsymbol{x}^{(t)}, \boldsymbol{x}^{(0)})}{p_{\boldsymbol{\theta}}(\boldsymbol{x}^{(t-1)} | \boldsymbol{x}^{(t)}, \boldsymbol{\eta})} + \log \frac{q_{\varphi}(\boldsymbol{\eta} | \boldsymbol{x}^{(0)})}{p_{\boldsymbol{\theta}}(\boldsymbol{\eta})} \right]
$$

$$
= E_q \left[ \sum_{t=2}^{T} D_{KL} \left( q(\boldsymbol{x}^{(t-1)} | \boldsymbol{x}^{(t)}, \boldsymbol{x}^{(0)}) || p_{\boldsymbol{\theta}}(\boldsymbol{x}^{(t-1)} | \boldsymbol{x}^{(t)}, \boldsymbol{\eta}) \right) - \log p_{\boldsymbol{\theta}}(\boldsymbol{x}^{(0)} | \boldsymbol{x}^{(1)}, \boldsymbol{\eta}) + D_{KL} \left( q_{\varphi}(\boldsymbol{\eta} | \boldsymbol{x}^{(0)}) || p_{\boldsymbol{\theta}}(\boldsymbol{\eta}) \right) \right]
$$

$$(22)$$

□

### B.1 PSEUDO-CODE OF DFITE

We present the diffusion model training algorithm in Algorithm 1, the sampling algorithm in Algorithm 2, and our ITE estimation algorithm in Algorithm 3. The statistics of the datasets are presented in Table 2.

---

**Algorithm 1:** Training

---

1 Indicate the observational data $\mathcal{X}$.
2 Initialize all the model parameters.
3 **while** *not converged* **do**
4     Sample $\boldsymbol{x}^{(0)} \sim \mathcal{X}$
5     Sample $\boldsymbol{\eta} \sim q_{\boldsymbol{\varphi}}(\boldsymbol{\eta}|\boldsymbol{x}^{(0)})$
6     Sample $t \sim \text{Uniform}(\{1,...,T\})$
7     Sample $\boldsymbol{x}_1^{(t)},...,\boldsymbol{x}_m^{(t)} \sim q(x^{(t)}|x^{(0)})$
8

$$L_\theta = \sum_{i=1}^m D_{KL}\left(q(\boldsymbol{x_i}^{(t-1)}|\boldsymbol{x}^{(t)},\boldsymbol{x_i}^{(0)})||p_{\boldsymbol{\theta}}(\boldsymbol{x_i}^{(t-1)}|\boldsymbol{x_i}^{(t)},\boldsymbol{\eta})\right)$$

$$L_\varphi = D_{KL}\left(q_{\boldsymbol{\varphi}}(\boldsymbol{\eta}|\boldsymbol{x}^{(0)})||p(\boldsymbol{\eta})\right)$$

9     Compute the gradients of the $L_\theta + \frac{1}{T}L_\varphi$
10     Perform the gradient descent.
11 **end**

---

**Algorithm 2:** Sampling

---

1 Sampling data points: $\boldsymbol{x}^{(T)} \sim \mathcal{N}(0,\boldsymbol{I})$.
2 **for** $t = T,...,1$ **do**
3     $\epsilon \sim \mathcal{N}(0,\boldsymbol{I})$ if $t > 0$, else $\epsilon = 0$
4     $x^{(t-1)} = \mu_{\boldsymbol{\theta}}(\boldsymbol{x}^{(t)},t,\eta) + \beta_t\epsilon$
5 **end**
6 return $\boldsymbol{x}^{(0)}$ as the unobserved confounders $z$

---

**Algorithm 3:** Learning algorithm of our model

---

1 Generating the unobserved confounders $z_1,...,z_m$ through Algorithm 2.
2 Indicate the observational data $(x_1,z_1,t_1,y_1),...,(x_m,z_m,t_m,y_m)$.
3 Indicate the scaling parameter $\alpha$ and $\beta$ .
4 Initialize all the model parameters.
5 Indicate the epoch number $E$.
6 Compute $u = \frac{1}{m}\sum_{i=1}^m t_i$.
7 Compute $w_i = \frac{t_i}{2u} + \frac{1-t_i}{2(1-u)}$ for $i = 1,...,m$
8 **for** *e in [0, E]* **do**
9     Sample mini-batch data $\mathcal{B}$ from $D$
10     Compute the gradients of the empirical loss:

$$g_1 = \nabla_W \frac{1}{|\mathcal{B}|}\sum_{i=1}^{|\mathcal{B}|} w_i L(y_i, f(\Phi(x_i,z_i),t_i))$$

11     Compute the gradients of the regularization:

$$g_2 = \nabla_W \beta \mathcal{R}(f)$$

12     Compute the gradients of the IPM term:

$$g_3 = \nabla_W \alpha IPM_G(\hat{p}_\Phi^{t=1}, \hat{p}_\Phi^{t=0})$$

13     Obtain the step size scalar $\rho$ with the Adam
14     Update the parameters:
$$W \leftarrow W - \rho(g_1 + g_2 + g_3)$$

15 **end**

---

