# OpenReview forum: "DFITE: Estimation of Individual Treatment Effect Using Diffusion Model"
_ICLR.cc/2024/Conference — Submitted to ICLR 2024_

### Official Review · Reviewer_otL9 · 2023-10-28

**Soundness:** 2 fair
**Presentation:** 2 fair
**Contribution:** 2 fair
**Rating:** 3
**Confidence:** 3

**Summary:**

This paper proposes to leverage diffusion models to generate unmeasured confounders in observational studies and obtain more accurate estimates of individual/heterogeneous treatment effects (ITEs). To this end, the authors derive the variational lower bound for the likelihood of the unobserved confounders, and devise a set of techniques to enable efficient training. The proposed methods are evaluated on 4 datasets, compared with 12 competing methods. Experiments show that the proposed methods achieve competitive performance.

**Strengths:**

1. Interesting direction and ideas.

This paper studies an interesting problem, which is whether advanced sampling tools such as diffusion models can be leveraged to improve understanding of causality. Also, estimating treatment effects in observational studies is an important problem, and the challenge of unmeasured confounding is significant. The idea of this paper may inspire future efforts in this direction.

2. Fruitful results.

This paper contains fruitful results in experiments and code-sharing. This is helpful for the progress of the community.

**Weaknesses:**

1. Lack of discussion on why diffusion models can tackle unmeasured confounding.

I think a huge missing part here is to justify why diffusion model can tackle causal inference problems and unmeasured confounding. This could be improved by re-organizing the paper a bit. Instead of directly coming up with diffusion model, it is helpful to first explain why generating synthetic unmeasured confounding can be useful, and how. Then, one can talk about why diffusion model is a useful tool in doing so. In particular, there appears to be no "causal" thinking across Section 3. I even cannot see what the unmeasured confounder is in the expressions. Is it $x^{(T)}$? Why can we learn it from data given that we never observe them? Is that because there are observable implications of a correct unmeasured confounder that we can leverage in the training? What does the variational lower bound mean? Why maximizing the log likelihood gives a good sampler for unmeasured confounding? Also, though section 4 appears at the end of theory & method, maybe it is more appropriate as a motivating point.

2. Missing details of the experiments.

Though the experiments are thorough, and given the provided codebase, I believe the methods are solid. But due to missing details I am confused how the performance is evaluated, such as how do we get the ground truth, and how is the ground truth defined.

3. Writing quality and clarity.

In general the quality of writing can be improved in this paper. Besides the above points on presentation, there are also many typos and broken sentences, which I point out in the Questions section.

**Questions:**

Most of my questions have been stated in the Weakness section. So my question below just provides more details.

- Why optimizing the loss function gives a good sampler for unmeasured confounding?

- What are the unmeasured confounders and ITEs in the experiments?

It seems that for ACIC and IHDP datasets, it is unclear whether there are unmeasured confounders. I'm also not sure how to access the true $\tau$ here. Do we have ground truths of the ITEs? Are they defined as $E[Y(1)-Y(0)|X]$ in the super-population, averaged over the randomness of unmeasured confounding conditional on $X$?


-----
Small typos & working issues:

1. The third line of the last paragraph of Section 1: "the likelihoodof"

2. I suggest using citep{} for references that do not function as a component of the sentence.

3. The sentence after equation (1): "Where..." shouldn't be a new sentence.

4. The sentence "At the previous time step, the kernel are responsible for adding noise to the points and at the next time step, which capable of modeling the distribution of points." in Section 3.1, after equation (2), seems broken.

---

### Official Review · Reviewer_RF2t · 2023-10-30

**Soundness:** 2 fair
**Presentation:** 2 fair
**Contribution:** 2 fair
**Rating:** 3
**Confidence:** 4

**Summary:**

The work aims to estimate ITE for datasets where the ignorability assumption does not hold. They propose to achieve this by using a diffusion model to generate the unobserved confounders. There are three main components in the proposed learning objective Eq (13): 1. Generate unobserved confounders via a forward-backward diffusion model, learn the functions that map (confounders, treatment) to the outcome, and balance treated and control distributions.

**Strengths:**

The authors demonstrate the effectiveness of their idea by comparing it with many baselines and on four datasets.

**Weaknesses:**

1. Writing could be improved. For example, for every single-line equation, the word "where" should not be capitalized.

**Questions:**

1. What exactly are the assumptions on the data $x, z, y, a$ are needed for the proposed methodology to work?

2. The random variables $x$ (observed confounders) and $z$ (unobserved confounders) follow different distributions. How can we use information of $x$, apply the diffusion model, and then infer the distribution of $z$?

3. Diffusion models were originally proposed for high-dimension data such as images. So what are the dimensions of data that the paper is generating and why a diffusion model is needed in the first place?

4. Can the authors explain what it means to "balance treated and control distributions"?

---

### Official Review · Reviewer_UsZZ · 2023-10-30

**Soundness:** 2 fair
**Presentation:** 1 poor
**Contribution:** 1 poor
**Rating:** 1
**Confidence:** 4

**Summary:**

This work leverages diffusion model to handle unmeasured confounder in causal inference. The intuition is to represent the unmeasured confounder using hidden variables.

### update: since there is no response at all, the reviewer decided to decrease the score.

**Strengths:**

Comprehensive numerical results are provided to support the proposed method

**Weaknesses:**

1. The writing is extremely poor, making the reviewer wondering whether the draft was proofread before submission:

- $\epsilon_{PEHE}$ and $\epsilon_{ATE}$ undefined before used in abstract
- in 2nd paragraph: "(e.g.,medicine)" --- missing space --- and "(e.g., demographic characteristics )" --- one extra space before ")".
- in 2nd paragraph: "DR", which the reviewer assumed to be "doubly robust", is again used before definition

It is not the reviewer's responsibility to proofread the paper and it is highly recommended that a careful revision is needed before resubmission.

2. Additionally, the review of existing work and motivation are unclear from the introduction:

"...and accordingly to infer the ITE in an unbiased settings..."
What is an "unbiased setting"? Does it mean the estimator is unbiased?

"VAEs make substantially weaker assumptions in generating the structure of the hidden confounders Louizos et al. (2017), which could restrict the model’s flexibility."
Why a weaker assumption leads to more restrictive method?

"For examples, GAN-based methods could be unstable in modeling ITE due to the adversarial losses. VAEs make substantially weaker assumptions in generating the structure of the hidden confounders Louizos et al. (2017), which could restrict the model’s flexibility. In order to address these challenges, in this paper, we propose to generate the unobserved confounders using diffusion model."

Based on the above review of existing work, it is very, very unclear why the proposed diffusion method can handle the issues (and not to mention that those issues are not stated clearly at all).

**Questions:**

See clarity questions above.

Moreover, in table 1:

How is the hyperparameter selected? what is the criterion? Indeed, there is recent work (this year) showing that the criterion for hyperparameter selection is even more important than the model itself. Please look at:

Machlanski, Damian, Spyridon Samothrakis, and Paul Clarke. "Hyperparameter tuning and model evaluation in causal effect estimation." arXiv preprint arXiv:2303.01412 (2023).

Curth, Alicia, and Mihaela van der Schaar. "In search of insights, not magic bullets: Towards demystification of the model selection dilemma in heterogeneous treatment effect estimation." arXiv preprint arXiv:2302.02923 (2023).

Wei, Song, et al. "Detecting Electricity Service Equity Issues with Transfer Counterfactual Learning on Large-Scale Outage Datasets." arXiv preprint arXiv:2310.03258 (2023).

If there were cross validation involved, please specify what error table 1 is reporting --- the current table cannot be compared to existing results in literature, e.g., table 1 in Dagonnet.

---

### Official Review · Reviewer_1VfB · 2023-10-31

**Soundness:** 3 good
**Presentation:** 2 fair
**Contribution:** 2 fair
**Rating:** 5
**Confidence:** 3

**Summary:**

In this paper, the authors aim to learn individualized treatment effects (ITE) from observational data with unobserved confounders. They propose to learn the unobserved confounder using a diffusion model and then incorporate it to estimate ITE. The authors examined the proposed model on synthetic and benchmark datasets, comparing it to existing methods for learning ITEs.

**Strengths:**

This paper considers an important problem in learning ITE when unobserved confounders exist. The novel part of the proposed model mainly lies in proposing the use of a diffusion model to infer the unobserved confounder based on observed variables.

**Weaknesses:**

The authors need to discuss when the unobserved confounders can and cannot be accurately inferred from the observed variables. Real data examples can be helpful to illustrate this and inform the readers when they encounter this problem in practice. Additionally, the rationale for using the diffusion model to infer the latent unobserved confounders and why the diffusion model could potentially offer an advantage compared to other methods needs more discussion in the Introduction section. Also, whether and why the diffusion model can recover the true unobserved confounder was not clear. Why this process offers more information than just using X in the ITE model was not clear either, as the unobserved latent confounder was solely inferred from X. Lastly, the experiment results do not fully justify the utility of learning the unobserved confounder for learning ITE and the benefit of the diffusion model. Detailed comments can be found in the questions.

**Questions:**

Can the authors provide more details on the rationale for using a diffusion model to generate the unobserved confounders? What benefit would it offer, and why is it appropriate?

In the first paragraph on page 2, I was not following the logic in the following sentence: “VAEs make substantially weaker assumptions in generating the structure of the hidden confounders Louizos et al. (2017), which could restrict the model’s flexibility.”

In the Introduction section, the authors give the example that the environment in which the patient lives and works (ita) can affect both the patient's socio-economic status (observed confounder X) and gene (the unobserved confounder Z). Can the authors use this example to illustrate the forward diffusion process and the reverse diffusion process? For all formulas in section 3.1, only ita and X are involved. How does Z come into play? It needed clarification on how z_i is derived by the diffusion model as zi ~ \miu_theta (c, t. ita_i) + beta_t \ipsilon (the third line after formula 13).

The authors use the Gaussian transition kernel. Would it only work when X is a continuous variable? Can X be more than one dimension? What if X is discrete?

Can the authors provide details on what variables are treated as X and Z in the two real data examples, ACIC 2016 and IHDP, and how are Z inferred?

Since Z is only inferred from X and both X and Z are included in the model to learn ITE. It was not clear to me how this additional step of inferring Z from X would help, especially when the ITE is learned from the flexible neural network models. Isn’t all the information already captured in X? Also, how the proposed diffusion model guarantee that Z is learned correctly?

As the authors pointed out, the distance metric (the last term in Formula 13) used to balance two distributions play a significant role in improving the estimation of ITE. Therefore, I wonder, is the diffusion model better than the VAE (used in CEVAE) to learn the unobserved confounder, or is the benefit mainly driven by the use of the distance metric? Also, even in the synthetic datasets, methods that do not account for unmeasured confounding such as TARnet and CFR_MMD have very good performance and even outperforms the proposed.

---

### Official Review · Reviewer_Jxsb · 2023-10-31

**Soundness:** 1 poor
**Presentation:** 3 good
**Contribution:** 2 fair
**Rating:** 3
**Confidence:** 4

**Summary:**

This work tackles the problem of unobserved confounder measuring in the scope of estimating Individual Treatment Effect (ITE). The authors proposed to use diffusion model to fit on the observed covariates and generate representations for unobserved covariates using sample from an intermediate time step in the reverse process. The generated representations are then incorporated in the downstream learning task, which is jointly optimized with the diffusion variational lower bound. Experiments are conducted on various datasets (including simulation datasets) with various benchmark, where the proposed model showed best performance among benchmarks on some of the evaluation metrics.

**Strengths:**

Paper is easy to follow, formulations and assumptions are clearly layed out. Algorithmic workflows are presented in detail. Experiments are done thoroughly with various benchmarks, and visualization of the results helped understanding model performance. Overall, the topic of the paper is of decent significancy as unobserved confounders is a crucial problem for causal identifiability, and incorporating modern deep learning tool to tackle this problem is an interesting angle.

**Weaknesses:**

I don't stand by the theoretical justification for using diffusion model to generate unobserved confounder. It does not align with the causal graph you proposed in Figure 1, in which $Z$ and $X$ are conditionally independent given $\eta$. Thus, generating $z$ by using $\mu_\theta$ plus noise makes no sense to me.

Besides, the factorization in the proof of the ELBO replies on the bayesian network for the generating process to be either $\eta -> x^{(0)} -> x^{(1)} ... -> x^{(T)}$ or $\eta <- x^{(0)} <- x^{(1)} ... <- x^{(T)}$, which is also unjustifiable to me.

The motivation for using diffusion model is also very doubtful. It is very unclear to me why one would use it to diffuse covariates vector, which in most real-world cases aren't even fully continuous, let alone having Gaussian noises, and in some case aren't even available in high-dimension. The simulation cases you proposed aren't really realistic to me for that reason. Although it is mentioned in one sentence in the introduction, it's still very much unclear to me why diffusion model is preferred over an VAE approach and why the assumption that the covariates are generated from a denoising process is preffered over no assumption.

In a bigger picture, on the scope of missing data imputation, prior work [1] has already proposed using diffusion model to impute missing data, and has advanced the "impute-then-generate" framework you're using to directly account for the uncertainty
of missing data in the learning process, and the drawback of "impute-then-generate" framework was illustrated.

Overall, the contribution of this paper is flawed and unsubstantial in my opinion.

[1] Ouyang, Yidong, et al. "MissDiff: Training Diffusion Models on Tabular Data with Missing Values." arXiv preprint arXiv:2307.00467 (2023).

**Questions:**

1. Could the authors elaborate on why diffusion model is preferred over an VAE approach for covariates imputation? In the introduction, you said VAE "could restrict the model’s flexibility". How is making less assumptions restricting model's flexibility? Isn't that the opposite?

2. What value does Theorem 1. add to the paper really? If there's any part of the proposed method that aids to identifiability or true latent recovery, please elaborate.

Misc.:
Line 3 of Eq 20: missing negative sign.

---

### Meta-Review · Area_Chair_kMAY · 2023-12-06

**Metareview:**

The paper introduces a novel approach to learning Individual Treatment Effects (ITE) from observational data with unobserved confounders, utilizing a diffusion model to capture the unobserved latent space.

pros:
- Addresses an important problem of learning ITE with unobserved confounders
- Provides comprehensive numerical results

cons:
- unclear motivation and theoretical justification for using the diffusion model to tackle unobserved confounding
- experiment results do not fully justify the utility of learning unobserved confounders
- lacking in writing quality

**Justification For Why Not Higher Score:**

unclear motivation and theoretical justification for using the diffusion model to tackle unobserved confounding

**Justification For Why Not Lower Score:**

NA

---

### Decision · Program_Chairs · 2024-01-16

Reject